# Anticancer Study of a Novel Pan-HDAC Inhibitor MPT0G236 in Colorectal Cancer Cells

**DOI:** 10.3390/ijms241612588

**Published:** 2023-08-09

**Authors:** Feng-Lung Tsai, Han-Li Huang, Mei-Jung Lai, Jing-Ping Liou, Shiow-Lin Pan, Chia-Ron Yang

**Affiliations:** 1School of Pharmacy, College of Medicine, National Taiwan University, Taipei 100, Taiwan; d07423201@ntu.edu.tw; 2TMU Research Center for Drug Discovery, Taipei Medical University, Taipei 110, Taiwan; hlhuang@tmu.edu.tw (H.-L.H.); mjl@tmu.edu.tw (M.-J.L.); jpl@tmu.edu.tw (J.-P.L.); slpan@tmu.edu.tw (S.-L.P.); 3Ph.D. Program in Drug Discovery and Development Industry, College of Pharmacy, Taipei Medical University, Taipei 110, Taiwan; 4School of Pharmacy, College of Pharmacy, Taipei Medical University, Taipei 110, Taiwan; 5Graduate Institute of Cancer Biology and Drug Discovery, College of Medical Science and Technology, Taipei Medical University, New Taipei City 235, Taiwan; 6TMU Research Center of Cancer Translational Medicine, Taipei Medical University, Taipei 110, Taiwan; 7Ph.D. Program for Cancer Molecular Biology and Drug Discovery, College of Medical Science and Technology, Taipei Medical University, New Taipei City 235, Taiwan

**Keywords:** colorectal cancer, HDAC inhibitors, cell cycle arrest, apoptosis

## Abstract

Colorectal cancer (CRC) is one of the most commonly diagnosed malignancies and a leading cause of cancer worldwide. Histone deacetylases (HDACs), which regulate cell proliferation and survival, are associated with the development and progression of cancer. Moreover, HDAC inhibitors are promising therapeutic targets, with five HDAC inhibitors approved for cancer treatment to date. However, their safety profile necessitates the exploration of well-tolerated HDAC inhibitors that can be used in cancer therapeutic strategies. In this study, the pan-HDAC inhibitor MPT0G236 reduced the viability and inhibited the proliferation of human colorectal cancer cells, and normal human umbilical vein endothelial cells (HUVECs) showed reduced sensitivity. These findings indicated that MPT0G236 specifically targeted malignant tumor cells. Notably, MPT0G236 significantly inhibited the activities of HDAC1, HDAC2, and HDAC3, Class I HDACs, as well as HDAC6, a Class IIb HDAC, at low nanomolar concentrations. Additionally, it promoted the accumulation of acetyl-α-tubulin and acetyl-histone H3 in HCT-116 and HT-29 cells in a concentration-dependent manner. Furthermore, MPT0G236 treatment induced G2/M cell cycle arrest in CRC cells by initially regulating the levels of cell-cycle-related proteins, such as p-MPM2; specifically reducing p-cdc2 (Y15), cyclin B1, and cdc25C levels; and subsequently inducing apoptosis through the caspase-dependent pathways and PARP activation. Our findings demonstrate that MPT0G236 exhibits significant anticancer activity in human colorectal cancer cells.

## 1. Introduction

Colorectal cancer (CRC) is the second leading cause of cancer-related deaths and the fourth most commonly diagnosed cancer worldwide [1]. Although traditional treatments, e.g., chemotherapy and surgery, have greatly increased the survival rate, adverse reactions induced by chemotherapy (e.g., pancytopenia, vomiting, and nausea) are the most significant obstacles that prevent patients from completing their treatment. More than one-half of CRC patients die due to recurrence or metastasis. The 5-year survival rate for late-stage CRC or metastatic patients remains as low as 10–15%, and metastatic CRC remains incurable for most patients [2,3]. Therefore, developing novel drugs to treat CRC remains a pressing need. With the progression of cancer research, different cancer hallmarks have been explored. For example, nonmutational epigenetic reprogramming is regarded as a cancer hallmark, and it can be explored as a potential therapeutic target [4]. Epigenetic modifications are thought to alter the structure between chromatin and DNA and control gene expression transcription without resulting in permanent changes to the DNA sequence [5]. They are regarded as important factors in the pathogenesis of various cancers, including CRC. Histone modification is an essential epigenetic alteration in CRC cells and is considered the most important driver of normal cells’ transformation to cancer cells [6,7]; the factors contributing to epigenetic alterations are commonly thought to be effective molecular targets in treatments.

Histone deacetylases (HDACs) play critical roles in epigenetic regulation through post-translational modification. The overexpression of HDACs in cancer patient samples was reported for different types of cancer, including colorectal cancer, and considerable evidence indicates that the overexpression of HDACs in patient colorectal cancer cells is associated with mortality [8,9]; on the other hand, gene knockout or the pharmacological inhibition of HDACs inhibited the cell growth of colon cancer cell models in vitro and in vivo [8,10,11,12]. Cell cycle arrest and apoptosis are critical mechanisms by which HDAC inhibitors selectively kill cancer cells. Cell cycle progression is mainly controlled by several core cell cycle regulatory proteins, including cyclin-dependent kinases (CDKs) and cyclins [13]. The overexpression of HDACs is associated with the deacetylation of histones, which regulates the tumor suppressor genes involved in cell cycle arrest. Studies indicated that HDACs regulate histone acetylation at the p21 promoter. The inhibition of HDACs leads to cell cycle arrest that is associated with an increase in p21 level and a decrease in cyclin A and cyclin B1 levels and induces apoptosis through the intrinsic pathway [14]. Caspase activation was shown to play an essential role with HDAC inhibitors to induce apoptosis. Caspase-3 is activated and regulated downstream of poly (ADP-ribose) polymerase (PARP), which is a critical apoptotic hallmark [15]. The substrates of HDACs are not limited to histones but include nonhistone proteins, including transcription factors such as p53; for example, HDAC inhibition induces p53 transcription and acetylation, leading to cell cycle arrest and caspase-dependent apoptosis [16]. HDAC6, a HDAC protein comprising an HDAC subclass, deacetylates its substrates tubulin and HSP90, which regulate protein folding and proteasomal degradation, respectively; specifically, the inhibition of HDAC6 induces growth arrest and apoptosis [17,18].

To date, five HDAC inhibitors (HDACIs) have been approved by the FDA [19]. Suberoylanilide hydroxamic acid (SAHA; vorinostat) was the first to be approved for treating cutaneous T-cell lymphoma in 2006. However, the safety profile of these HDACIs shows potentially adverse effects such as myelosuppression, cardiac toxicity, and drug-drug interactions mediated through the cytochrome P450 system, which limit their use [20,21]. Further, the challenge of HDACIs treatment is the need for progress in targeting solid tumors, which is currently being explored [19]. Safe and well-tolerated HDACIs need to be evaluated in a cancer therapeutic strategy. Our previous study indicated that MPT0G236 exhibits potent proliferation inhibition in human solid tumor cancer cells. In vivo data revealed that the compound significantly reduced HCT-116 human CRC tumor volume without affecting body weight [22]. Hence, in the present study, we aimed to explore the possibility of applying MPT0G236 to HDAC inhibition and evaluate its anticancer effect. Moreover, we elucidated the underlying mechanisms of its action using human colorectal cancer cells.

## 2. Results

### 2.1. Antiproliferative Effect of MPT0G236 on Cancer Cells

Our previous study indicated that MPT0G236 significantly inhibited cell proliferation in three types of solid cancer cells, and the GI_50_ values were 0.08 ± 0.001 µM in HCT-116 cells, 0.15 ± 0.01 µM in PC-3 cells, and 0.27 ± 0.02 µM in A549 cells [22]. All the values were lower than those of SAHA, suggesting that MPT0G236 potently inhibited the proliferation of cancer cells [22]. The in vivo results also demonstrated that oral administration of 50 mg/kg and 100 mg/kg MPT0G236 markedly suppressed HCT-116 tumor growth by 32.2% and 58.8%, respectively [22]. Therefore, in this study, we selected HCT-116 colorectal cancer cells, as in the previous study, and included another colorectal cancer cell, HT-29, to evaluate the antiproliferative effect of MPT0G236 on CRC cells. The cytotoxic effect of MPT0G236 was assessed first in the HCT-116 and HT-29 cells. As shown, MPT0G236 reduced the viability of both cells in a concentration-dependent manner, with IC_50_ values of 0.35 μM and 0.88 μM for the HCT-116 and HT-29 cells, respectively (Figure 1A). We then examined whether MPT0G236 inhibited the growth of the HCT-116 and HT-29 cells. The results showed that the GI_50_ values of MPT0G236 were 0.10 μM for the HCT-116 cells and 0.31 μM for the HT-29 cells, with both of these levels lower than those of SAHA in these cells (Figure 1B). Furthermore, we measured the cytotoxic effect of MPT0G236 on normal human umbilical vein endothelial cells (HUVECs). The results showed that after MPT0G236 treatment, the percentage of viable normal cells was higher than that of CRC cells, and, with an IC_50_ of 15.14 μM, normal cells were more than 10-fold less sensitive to MPT0G236 treatment than colorectal cancer cells (Figure 1C). These findings suggest that MPT0G236 specifically targets cancer cells and exhibits broad safety.

### 2.2. HDAC Activity Inhibited by MPT0G236

We next investigated the inhibitory potency and selectivity of MPT0G236 for HDAC isoforms using SAHA as the reference compound. As shown in Table 1, the results demonstrated that MPT0G236 significantly inhibited the activity of HDAC1 (IC_50_ = 14 nM), HDAC2 (IC_50_ = 11.5 nM), and HDAC3 (IC_50_ = 70 nM), which are Class I HDACs, as well as HDAC6 (IC_50_ = 15 nM), which is a Class IIb HDAC, at low nanomolar concentrations. The inhibitory effect of MPT0G236 on HDAC activity was similar to that of SAHA; however, MPT0G236 exhibited greater potency than SAHA, with lower IC_50_ values against recombinant HDACs from Classes I/IIb. These results suggest that MPT0G236 is a potent pan-HDAC inhibitor that particularly targets Class I/IIb HDACs.

### 2.3. MPT0G236 Increased α-Tubulin and Histone H3 Acetylation Abundance in CRC Cells

The structure of MPT0G236 is shown in Figure 2A. Since histone H3 and α-tubulin are downstream targets of HDAC Class I and HDAC6, respectively [23], we evaluated the inhibitory effect of MPT0G236 on HDACs by examining increases in the acetylation of each protein in a concentration-dependent manner. As shown in Figure 2B, MPT0G236 increased the acetylation of α-tubulin and histone H3. In HCT-116 cells, treatment with 0.3 or 1 μM MPT0G236 significantly increased the accumulation of acetyl-α-tubulin and histone acetylation, respectively. In HT-29 cells, treatment with 1 μM MPT0G236 significantly elevated histone acetylation, and treatment with 3 μM MPT0G236 had the same effect on acetyl-α-tubulin marks (Figure 2B). Compared with SAHA, MPT0G236 exhibited more potent inhibitory effects on both colorectal cancer cells. These results were consistent with the inhibitory effects of MPT0G236 on Class I HDACs and Class IIb HDAC6, as shown in Table 1.

### 2.4. Effects of MPT0G236 on Mitotic Arrest in HCT-116 and HT-29 Cells

To evaluate cell cycle progression after treatment with MPT0G236, we treated both cells with MPT0G236 and SAHA at a concentration of 3 μM and cultured the cells for 16, 24, and 48 h (Figure 3), and we treated cells at different concentrations of MPT0G236 and cultured them for 24 h and 48 h (Figure 4); cellular DNA was stained with propidium iodide, and then the DNA content was measured by flow cytometry. The results of cell cycle distribution showed that after treatment with MPT0G236, the proportion of cells in the G2/M phase increased after 16 and 24 h of treatment and decreased 48 h after treatment, compared with the levels in the control group of both cells (Figure 3A,B); additionally, treatment with MPT0G236 caused a significant increase in the number of HCT-116 cells in the sub-G1 phase from 16 to 48 h after treatment and in HT-29 cells from 24 to 48 h after treatment (Figure 3A,B). As shown in Figure 4A, HCT-116 cells were treated with different concentrations of MPT0G236 for 24 h, and G2/M arrest was observed at concentrations that ranged from 0.3 to 3 μM; then, the number of cells in G2/M arrest decreased, subsequently causing a clear proportion of cells in the sub-G1 phase to be sustained at an increased level until 48 h after treatment. In HT-29 cells, G2/M arrest was observed only 24 h after treatment with a high concentration of MPT0G236 (3 μM), which led to an increase in the proportion of cells in the sub-G1 phase (Figure 4B). Compared with SAHA, MPT0G236 showed similar G2/M arrest levels in HCT-116 cells (Figure 4A) but exerted a more significant arresting effect in HT-29 cells (Figure 4B). We further evaluated the mechanism by which MPT0G236 treatment induced cell cycle arrest in the G2/M phase. Phospho-cdc2 (Y15), cdc25C, and cyclin B1 are critical regulatory proteins in the G2/M phase [24]. Cdc2 is a crucial protein kinase for regulating the G2/M transition; it forms a complex with cyclin B1 to become activated and initiate mitosis. However, before this complex can be activated, cdc2 must be phosphorylated at the Y15 residue, which inhibits its activity. Cdc25C is a phosphatase that removes the inhibitory phosphate group from p-cdc2 (Y15), inducing cdc2 activation and, thus, initiating mitosis [24]. The results of this experiment indicated that the levels of cdc25C, p-cdc2 (Y15), and cyclin B1 were decreased in response to MPT0G236 treatment (>0.3 μM) in HCT-116 cells (Figure 5); however, compound treatment in HT-29 cells caused cdc2 (Y15) phosphorylation only at high concentrations (>1 μM) and did not change cyclin B1 or cdc25C expression (Figure 5). In addition, p-MPM2 is closely related to the M phase of the cell cycle; its increase is considered a marker of cells undergoing mitotic arrest [25]. MPT0G236 did not change the level of p-MPM2 in HCT-116 cells but increased its level in HT-29 cells (Figure 5). These results indicated that MPT0G236 blocked the cell cycle at the G2 phase in HCT-116 cells and the M phase in HT-29 cells by modulating the activity of the relevant transition proteins.

### 2.5. MPT0G236 Induced Apoptosis in Human Colorectal Cancer Cells

The sub-G1 fraction of the HCT-116 cells in the cell cycle distribution assay was markedly increased after 16 h of treatment with MPT0G236 and continued to increase after 48 h when the concentration was 3 μM (Figure 3A). In the HT-29 cells, an increase in the sub-G1 proportion caused by MPT0G236 was observed at high concentrations (>1 μM) and over a more extended treatment period (>24 h), and the increase in the sub-G1 cell proportion was less pronounced than that of the HCT-116 cells (Figure 4B). SAHA showed a weaker effect on both cells (Figure 3 and Figure 4). Furthermore, we evaluated apoptosis-related proteins, including PARP and the activated forms of caspase 3, caspase 8, and caspase 9, which are hallmark proteins of apoptosis and induce cell apoptosis through both the extrinsic and intrinsic pathways. The results showed that treatment with MPT0G236 increased the levels of the cleaved forms of PARP, caspase 3, caspase 8, and caspase 9 in both cells after 24 h (Figure 6A,B), and the increase in cleaved PARP and the caspase 3 level persisted up to 48 h (Figure 6A,B). In contrast, SAHA exhibited a less pronounced increase in the activated forms of PARP, caspase 3, and caspase 8 when administered at the same concentration as MPT0G236 (Figure 6A,B). These results suggest that MPT0G236 triggers apoptosis in HCT-116 and HT-29 cells, and this apoptotic effect is mediated through a caspase-dependent pathway.

## 3. Discussion

The FDA approved several HDAC inhibitors for various hematological cancers; however, due to the low potency and undesirable safety profiles of these inhibitors, HDACIs that offer greater efficacy, safety, and tolerability need to be identified [20,21]. Furthermore, the effects of HDAC inhibitors on solid tumors remain to be investigated. In our previous study, MPT0G236 demonstrated remarkable enzymatic and inhibitory activity in cells. MPT0G236 exhibited cytotoxicity against human solid tumor cancer cells, including HCT-116, PC-3, and A549 cells [22]. In this study, we evaluated HCT-116 and HT-29 colorectal cancer cells, which showed high sensitivity to MPT0G236, whereas the growth of normal cells was unaffected. We also demonstrated that MPT0G236 inhibited the activity of Class I HDACs (HDAC1, HDAC2, HDAC3, and HDAC8), as well as of a Class IIb HDAC, HDAC6, leading to the increased acetylation of α-tubulin and histone H3 in both colorectal cancer cells. MPT0G236 displayed an inhibitory pattern similar to that of SAHA but exhibited higher potency, as indicated by a lower IC_50_ value. The greater inhibition of MPT0G236 compared to SAHA may be attributed to its unique structural features. While SAHA contains an aliphatic alkane acting as the linker between the hydroxamic acid and phenylamide. MPT0G236 is composed of three core components: a heterocycle, a benzenesulfonyl group, and an *N*-hydroxyacrylamide moiety. Previous studies indicated that compounds exhibiting these structures displayed stronger anti-proliferative activities than SAHA [22]. In the development of MPT0G236, the previous study incorporated these three components but introduced crucial modifications by adding a sulfonyl linker to connect the heterocycle and the benzene group, and the *N*-hydroxyacrylamide group was added at the meta-carbon position of the benzene ring [22]. This unique structure plays a significant role in the biological activity of MPT0G236, contributing to its enhanced potency compared to SAHA.

Studies indicated that HDACIs selectively induce cell cycle arrest and apoptosis in cancer-transformed cell lines without affecting normal cells [26,27]. Normal cells rely on histone modifications to maintain proper chromatin structure, and HDACs play a critical role in modulating the machinery associated with these modifications. However, HDAC dysregulation is common in many tumor types [28]. Analyses of tumor tissues from patients showed that Class I HDACs are dysregulated and overexpressed in several solid tumors, including breast [29], prostate [30], gastric [31], lung [32], hepatocellular [33], and colorectal cancers [9,34]. The high expression of Class I HDACs correlates with the advanced-stage disease and poor survival of patients with certain cancers, including colorectal cancer [34]. Knockdown of Class I HDACs leads to the upregulation of p21, resulting in the inhibition of cyclin-dependent kinase activity, cell cycle arrest, and an increase in the number of cells undergoing apoptosis [35,36].

Histone deacetylase inhibitors were extensively studied, showing profound cytotoxic effects by inducing cell cycle arrest and apoptosis [37]. In addition to apoptosis induction, the inhibition of HDACs and the subsequent increase in histone acetylation mark abundance were associated with cell cycle arrest mediated through the upregulation of the tumor suppressor protein p21, which is encoded by the *CDKN1A* gene and functions as a cyclin-dependent kinase inhibitor [38]. Numerous studies reported this effect of HDAC inhibition. For instance, quisinostat, an HDAC Class I inhibitor, increased p53 acetylation and upregulated p21 expression, causing G1 phase arrest and inducing mitochondria-mediated cell apoptosis [39]. Trichostatin A (TSA) and SK-7041 caused G2/M cell cycle arrest by upregulating p21 and downregulating cyclin B1, the anti-apoptotic protein Mcl-1, and Bcl-XL in pancreatic cancer cell lines [40]. Romidepsin and SAHA increased histone and tubulin acetylation, downregulated the expression of cdc2 and cyclin B1, reduced the level of phosphorylated Rb (p-Rb), and increased p21 expression, and these changes led to G2/M arrest in breast and lung cancer cell lines [41]. By investigating the mechanism of action of MPT0G236, we demonstrated that the compound induced G2/M cell cycle arrest and increased the sub-G1 cell subpopulation (Figure 3 and Figure 4). Moreover, MPT0G236 treatment altered the expression of G2/M transition regulatory proteins, resulting in G2 arrest in HCT-116 cells by decreasing the levels of p-cdc2 (Y15), cdc25C, and cyclin B1. Additionally, MPT0G236 treatment increased the levels of mitosis-specific p-MPM2 proteins in HT-29 cells, indicating mitotic arrest (Figure 5). These results suggest that MPT0G236 mediates a different checkpoint arrest pattern in HCT-116 and HT-29 cells.

Research showed that HDAC inhibitors can cause cell cycle arrest in different phases depending on the specific HDAC inhibitor and cell type. Due to their distinct genetic features, HCT-116 and HT-29 cells are commonly used in studies. HCT-116 cells carry mutated KRAS and PIK3CA and exhibit microsatellite instability, while HT-29 cells carry mutated TP53, BRAF, and PIK3CA and exhibit microsatellite stability [42]. Although both cells exhibit cell cycle arrest and growth inhibition after treatment with HDAC inhibitors such as SAHA and panobinostat, microarray profiling revealed that these HDAC inhibitors induce cell-specific changes in gene expression involved in mitosis [43]. For example, aurora kinase B, which plays a role in centrosome function and mitotic spindle formation, was shown to be downregulated by both SAHA and panobinostat in HCT-116 cells but not in HT-29 cells [43].

Apoptosis is mediated through two pathways: the extrinsic and intrinsic pathways. Caspase 9 and caspase 8 are the initiator caspases for the intrinsic and extrinsic pathways, respectively. Both these pathways activate the common downstream proteins caspase 3 and PARP [44]. In our study, we demonstrated that MPT0G236 inhibited the activity of HDAC1, HDAC2, HDAC3, and HDAC8 (Class I), as well as HDAC6 (Class IIb) (Table 1). Inhibition of these HDACs commonly induces apoptosis through both the extrinsic and intrinsic pathways, which is a common mechanism for inducing cytotoxicity in cancer cells. For example, Chen et al. demonstrated that a novel pan-HDAC inhibitor activated the caspase cascade, leading to an increase in the expression of the proapoptotic protein Bim and a decrease in the expression of the antiapoptotic proteins Bcl-2 and Bcl-xL and the FLICE-inhibitor protein [10]. Henderson et al. revealed that TSA, a pan-HDAC inhibitor, induced apoptosis through caspase 9 activation, which depends on p53 activity regulation in breast cancer MCF-7 cells [45]. Lee et al. demonstrated that a novel HDAC1/2 inhibitor activated apoptosis in DLD1 and HCT-116 cells through the activation of cleaved PARP and BAK, modulating cell cycle distribution and increasing p21 protein levels [46]. SAHA and several chimeric HDAC inhibitors led to a significant increase in the apoptosis rate, as determined by the measurements of the levels of the activated effector caspases 3 and 7 in HCT-116 cells [47]. HDAC6 mediates and coordinates the major pathways involved in the degradation of misfolded and aggregated proteins, which are functions dependent on molecular chaperones that were demonstrated to play an essential role in promoting apoptosis or cell cycle arrest in human cancer cells [18,27]. A dual HDAC6/HSP90 inhibitor inhibited the cell growth of lung cancer cell lines in association with caspase-dependent apoptosis, further suppressing the growth in human H1975 xenograft tumors [48]. AR-42, a dual Class I and Class IIb HDAC inhibitor, induced caspase-mediated apoptotic death in SW-620 colorectal cancer cells and potently inhibited SW-620 xenograft growth in SCID mice [49].

Our results indicated that MPT0G236 treatment increased the number of cells in both lineages in the sub-G1 phase (Figure 3 and Figure 4) and induced apoptosis through a caspase-dependent pathway, which included caspase 8 and caspase 9 activation, further increasing the levels of the cleaved forms of caspase 3 and PARP (Figure 6). Overall, MPT0G236 showed pan-HDAC inhibition, potentially affecting a broader range of cellular processes and leading to apoptosis and cell cycle arrest.

## 4. Materials and Methods

### 4.1. Materials

MPT0G236 and SAHA (vorinostat) were synthesized by Jing-Ping Liou (School of Pharmacy, College of Pharmacy, Taipei Medical University, Taipei, Taiwan) at a purity exceeding 98%; the product was previously characterized [22]. MTT (3-(4,5-dimethylthiazol-2-yl)-2,5-diphenyltetrazolium bromide), sulforhodamine B (SRB), and propidium iodide (PI) were purchased from Sigma Chemical Co. (St. Louis, MO, USA); primary antibodies against acetyl-histone H3, acetyl-α-tubulin, PARP, cleaved caspase 3, caspase 8, caspase 9, cyclin B1, cdc25C, and phosphorylated cdc2 Tyr15 (p-cdc2 (Y15)) were purchased from Cell Signaling Technologies (Beverly, MA, USA); phosphorylated mitotic protein monoclonal 2 (p-MPM2) was purchased from Millipore (Bedford, MA, USA); β-actin was purchased from ABclonal (Woburn, MA, USA); HRP-conjugated anti-mouse and anti-rabbit IgG secondary antibodies were obtained from Jackson ImmunoResearch Inc. (West Grove, PA, USA).

### 4.2. Cell Culture

The HCT-116 and HT-29 human colorectal cancer cells and human umbilical vein endothelial cells (HUVECs) were purchased from American Type Culture Collection (ATCC, Manassas, VA, USA). The HCT-116 and HT-29 cells were cultured in McCoy’s 5A medium containing 10% (*v*/*v*) fetal bovine serum (FBS), penicillin (100 units/mL), streptomycin (100 μg/mL), and amphotericin B (0.25 mg/mL). The HUVEC cells were maintained in M199 with 20% FBS, penicillin (100 units/mL), streptomycin (100 μg/mL), and amphotericin B (0.25 mg/mL). The cells were maintained in an environment with humidified air and 5% CO_2_ at 37 °C and subcultured when the cell density reached 90% in a dish.

### 4.3. HDAC Enzyme Inhibition Assay

HDAC activity inhibition assays were performed by Reaction Biology Corp. (Malvern, PA, USA). Compounds were dissolved in DMSO and evaluated on the basis of the 10-dose IC_50_, at least, with a 3-fold dilution series starting at 10 μM.

### 4.4. Cell Proliferation Assay

Cells were seeded in 96-well plates at a density of 5000 cells per well and incubated overnight to allow adhesion. The basal cells were fixed with 10% (*w*/*v*) trichloroacetic acid for 10 min. The remaining cells were treated with MPT0G236 or SAHA at different concentrations for 48 h. After treatment, the cells were fixed with 10% TCA, stained with SRB at a concentration of 0.4% (*w*/*v*), and washed with 1% acetic acid. After drying, the stained cells were lysed with 10 mM Trizma base, and the absorbance was measured at a wavelength of 515 nm. The concentration of the evaluated compounds that reduced cell proliferation by 50% (GI_50_) was calculated.

### 4.5. Cell Viability Assay

Cells were seeded in 96-well plates at a density of 5000 cells per well and incubated overnight to allow adhesion. The medium was replaced with fresh medium, followed by the addition of MPT0G236 or SAHA at various concentrations for 48 h treatment. After treatment, an MTT solution (1 mg/mL) was added to the cell culture, and the plates were incubated at 37 °C for an additional 2 h. The MTT solution was then removed, the cells were lysed with 100 µL of dimethyl sulfoxide (DMSO) added to each well, and the absorbance was read at a wavelength of 550 nm. The concentrations at which the analyzed compounds induced inhibition by 50% (IC_50_) were calculated by using the GraphPad Prism 9 version 9.5.1 (528) program based on a sigmoidal dose-response equation.

### 4.6. Flow Cytometry

Cells were treated with MPT0G236 or SAHA to determine the cell cycle distribution after treatment with different concentrations and durations. The cells were harvested by trypsinization and washed with phosphate-buffered saline. The cells were pelleted, resuspended, and fixed in 75% ethanol (*v*/*v*) at −20 °C overnight. The ethanol was removed after centrifugation at 400× *g* for 3 min at 4 °C. The pellets were incubated with 0.2 mL of a phosphate–citric acid buffer (0.2 M Na_2_HPO_4_ and 0.1 M citric acid; pH 7.8) for 20 min at room temperature. The cells were then stained with propidium iodide staining buffer containing Triton X-100 (0.1% *v*/*v*), RNase A (100 µg/mL), and propidium iodide (80 µg/mL) for 30 min in a dark room. The cell cycle distribution was analyzed by FACScan flow cytometry using BD AccuriTM C6 Flow cytometer and software (Becton Dickinson, Mountain View, CA, USA).

### 4.7. Immunoblot Analyses

Cells were washed with PBS and incubated with lysis buffer (20 mM HEPES, pH 7.4; 2 mM EGTA; 0.1% Triton X-100; 50 mM β-glycerophosphate; 1 mM DTT; 10% glycerol; 1 μg/mL leupeptin; 1 mM sodium orthovanadate; 1 mM phenylmethylsulfonyl fluoride; 5 μg/mL aprotinin) for 10 min at 4 °C. The cells were then scraped off and incubated on ice for an additional 10 min before being centrifuged at 17,000× *g* for 30 min at 4 °C. The total protein volume was determined, and equal amounts of proteins were separated by electroporation on sodium dodecyl sulfate polyacrylamide gels (SDS-PAGE) and transferred onto a nitrocellulose membrane. The membrane was blocked with 5% (*v*/*v*) fat-free milk in Tris-buffered saline containing 0.1% Tween 20 (TBST) for 1 h at room temperature and then incubated with primary antibodies in TBST overnight at 4 °C. After washing three times, the membranes were incubated with HRP-conjugated secondary antibodies for 1 h at room temperature. Signals were detected with chemiluminescence reagents (GE Healthcare Corp., Buckinghamshire, UK), and the membranes were scanned using a ChemiDocTM MP Imaging System. Protein levels in the Western blots were then quantified using ImageJ software V1.53k.

### 4.8. Statistical Analysis

Each experiment was repeated at least three times, and the data are presented as the mean ± SEM. One-way ANOVA was performed to analyze data. When significant intergroup differences were found, Tukey’s post hoc test was performed to determine the pairs of groups with statistically significant differences. A *p* value less than 0.05 was considered to be statistically significant.

## 5. Conclusions

Our results showed that MPT0G236 is a novel pan-HDAC inhibitor that displayed a meaningful antiproliferative effect in colorectal cancer cells that resulted in cell cycle G2/M arrest and caspase-dependent apoptosis. This anticancer effect of MPT0G236 was greater than that of SAHA and explicitly targeted cancer cells. Our previous study revealed that this compound significantly reduced HCT-116 tumor volume in vivo without affecting body weight [22]. The results from the present demonstrated that MPT0G236 shows the potential to be a novel drug target in colorectal cancer treatment.

## Figures and Tables

**Figure 1 ijms-24-12588-f001:**
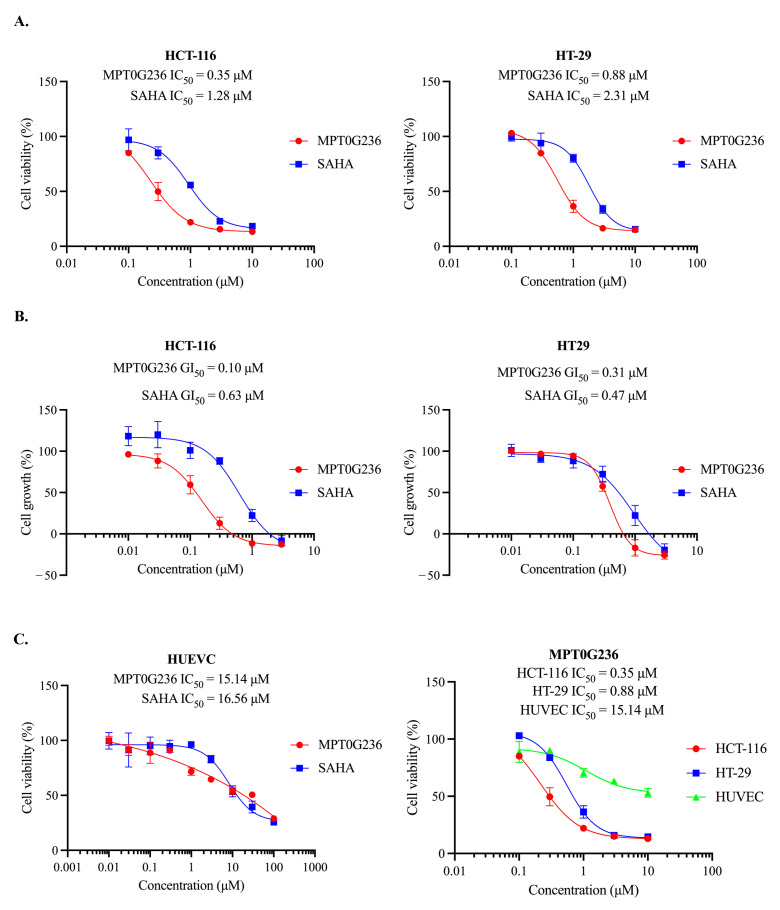
MPT0G236 exhibited inhibitory effects by reducing the viability and proliferation rate of human CRC cells. The impacts of MPT0G236 on cell viability (**A**,**C**) and cell growth (**B**) were evaluated with HCT-116 and HT-29 human colorectal cancer cells, as well as human umbilical vein endothelial cells (HUVECs). The cells were treated with different concentrations of the compound for 48 h. Cell viability was measured using an MTT assay, while cell growth was assessed using an SRB assay. The data are shown as the mean ± SEM on the basis of three independent experiments.

**Figure 2 ijms-24-12588-f002:**
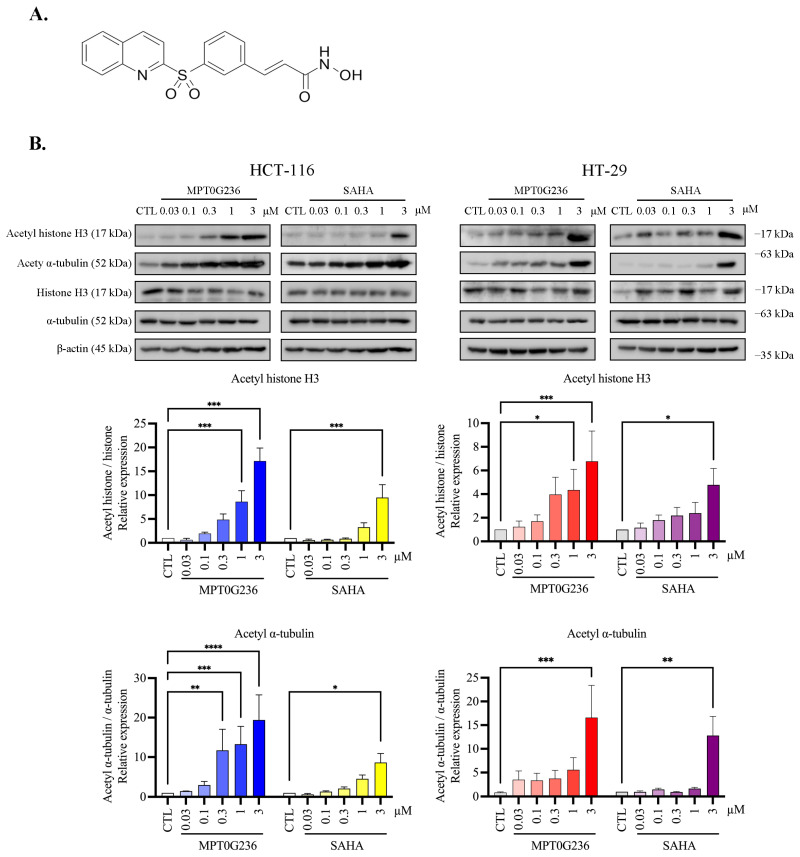
MPT0G236 induced acetylation of α-tubulin and histone H3 in human colorectal cells. (**A**) The structure of MPT0G236. (**B**) HCT-116 and HT-29 cells were incubated with MPT0G236 or SAHA for 24 h; cells were lysed, and the proteins in the cell extracts were analyzed by Western blotting. Protein expression was quantified by using the computerized image analysis application in ImageJ software V1.53k. The data are shown as the mean ± SEM on the basis of at least three independent experiments. * *p* < 0.05, ** *p* < 0.01, *** *p* < 0.001 and **** *p* < 0.0001 compared with the control group.

**Figure 3 ijms-24-12588-f003:**
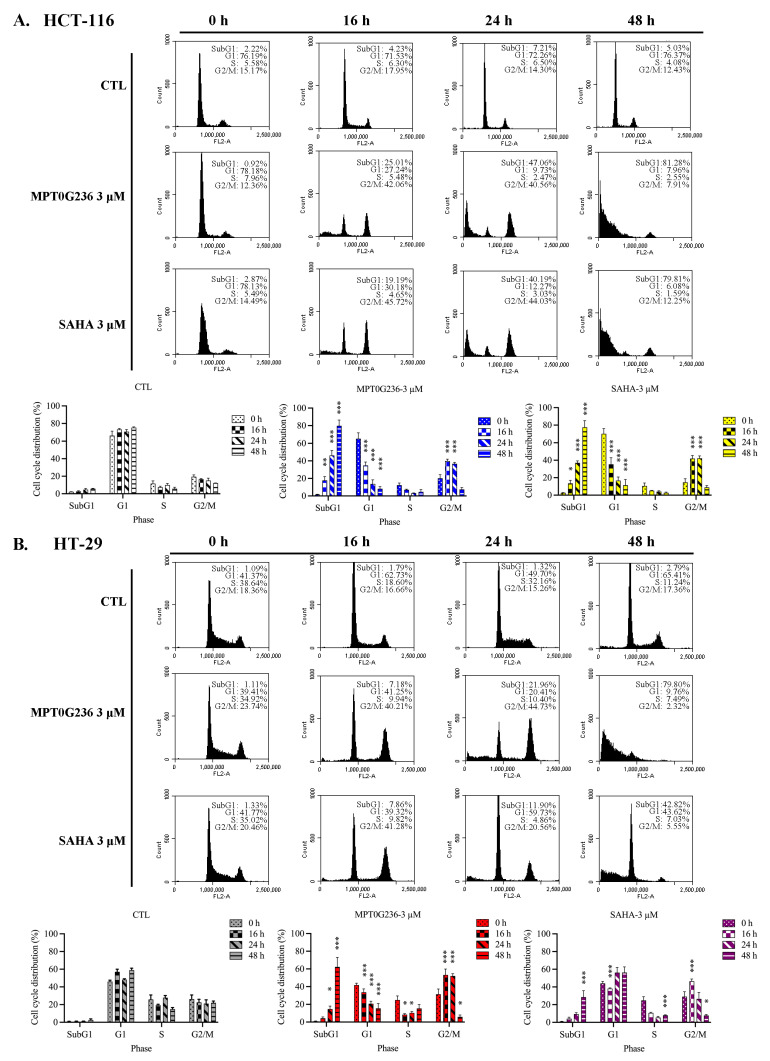
The change in the number of human CRC cells in each cell cycle phase after MPT0G236 treatment. (**A**) HCT-116 and (**B**) HT-29 cells were treated with 3 μM MPT0G236 or SAHA for 16, 24, and 48 h. The cell cycle distribution was measured using flow cytometry. The data are shown as the mean ± SEM on the basis of three independent experiments. * *p* < 0.05, ** *p* < 0.01, and *** *p* < 0.001 compared with the control group.

**Figure 4 ijms-24-12588-f004:**
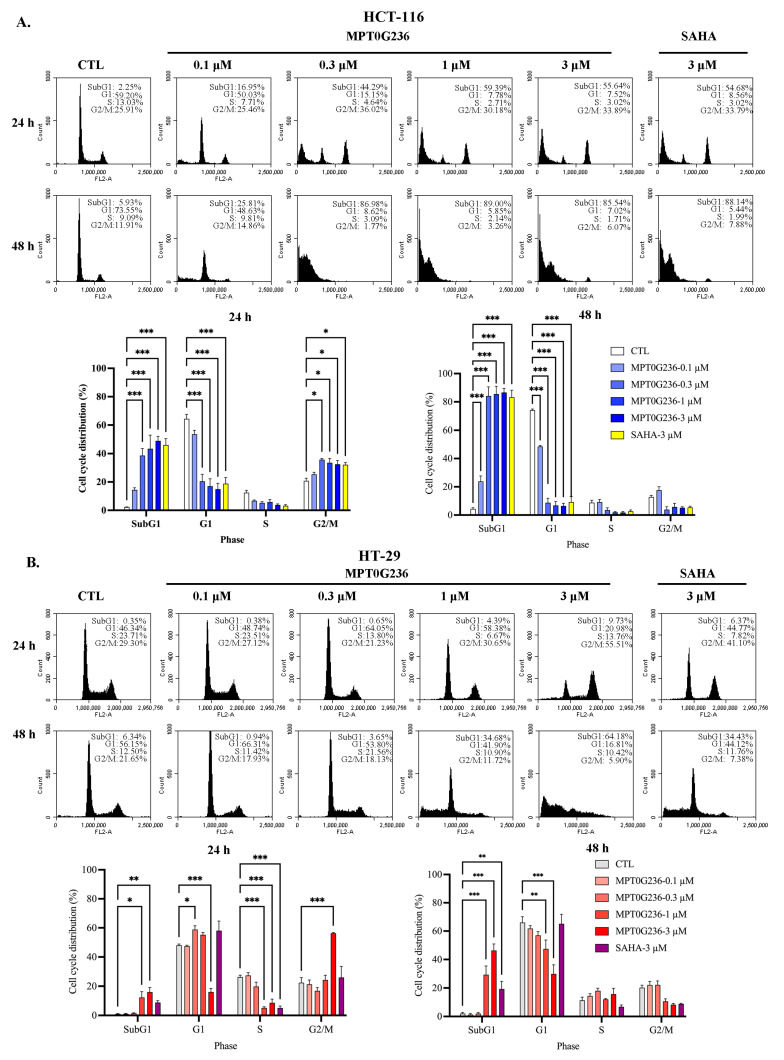
MPT0G236 changed the cell cycle distribution of HCT-116 and HT-29 cells. (**A**) HCT-116 and (**B**) HT-29 human colorectal cancer cells were treated with the indicated concentrations of MPT0G236 or 3 μM SAHA for 24 or 48 h. The cell cycle distribution was measured using flow cytometry. The data are shown as the mean ± SEM on the basis of three independent experiments. * *p* < 0.05, ** *p* < 0.01, and *** *p* < 0.001 compared with the control group.

**Figure 5 ijms-24-12588-f005:**
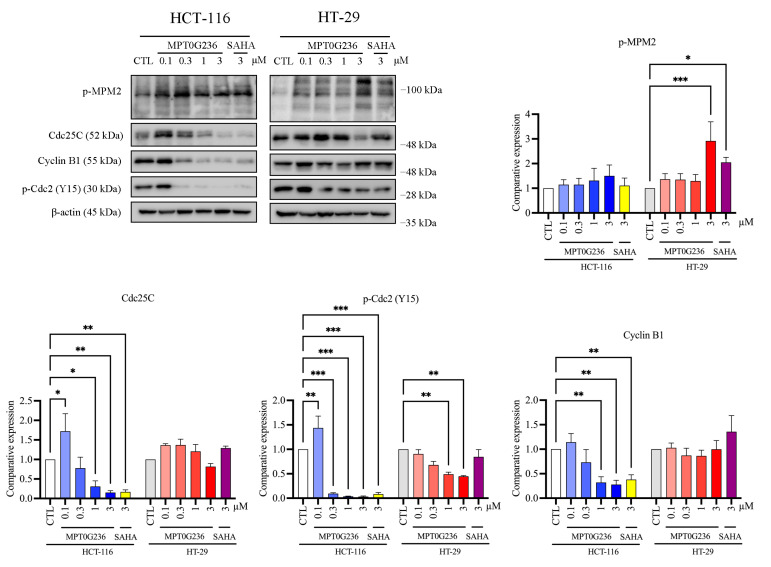
MPT0G236 changed the levels of a protein involved in the G2/M transition. HCT-116 or HT-29 cells were incubated with or without the indicated concentration of MPT0G236 or 3 μM SAHA for 24 h. Whole-cell lysates were subjected to Western blotting using the specified antibodies. Protein expression was quantified using the computerized image analysis application in ImageJ software V1.53k. The data are shown as the mean ± SEM on the basis of three independent experiments. * *p* < 0.05, ** *p* < 0.01, and *** *p* < 0.001 compared with the indicated group.

**Figure 6 ijms-24-12588-f006:**
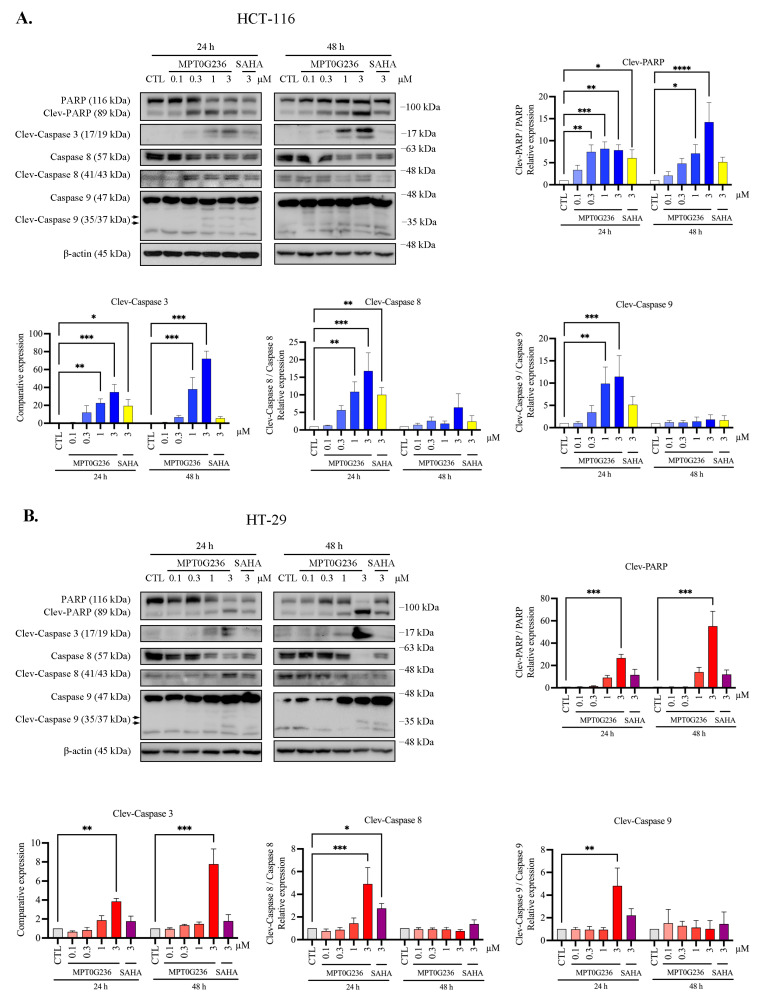
MPT0G236 changed the levels of proteins that regulate apoptosis. HCT-116 (**A**) and HT-29 (**B**) cells were incubated for 24 and 48 h with or without the indicated concentrations of MPT0G236 or 3 μM SAHA. Whole-cell lysates were subjected to Western blotting with the indicated antibodies. Protein expression was quantified by using the computerized image analysis application in ImageJ software V1.53k. The data are shown as the mean ± SEM on the basis of three independent experiments. * *p* < 0.05, ** *p* < 0.01, *** *p* < 0.001 and **** *p* < 0.0001 compared with the indicated group.

**Table 1 ijms-24-12588-t001:** HDACs’ inhibitory activity (IC_50_) of MPT0G236 and SAHA.

HDAC Isoforms (IC_50_ (nM))									
Class	Class I				Class IIa				Class IIb
HDAC Isoform	HDAC1	HDAC2	HDAC3	HDAC8	HDAC4	HDAC5	HDAC7	HDAC9	HDAC6
MPT0G236	14 ^a^	11.5 ^a^	70	130 ^a^	19,650	15,500	12,300	12,600	15 ^a^
SAHA ^b^	110 ^a^	120 ^a^	64.3 ^b^	306 ^b^	76,000 ^b^	27,200 ^b^	105,000 ^b^	141,000 ^b^	110 ^a^

^a^ Value obtained from [22]. ^b^ Value obtained from Reaction Biology Corporation. Retrieved from https://www.reactionbiology.com/services/target-specific-assays/epigenetic-assays/hdac-assays, accessed on 1 June 2023.

## Data Availability

Not applicable.

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
