# Peer review of "Anticancer Study of a Novel Pan-HDAC Inhibitor MPT0G236 in Colorectal Cancer Cells"

_ijms, 2023, doi:10.3390/ijms241612588_

Round 1
Reviewer 1 Report
The author delivered a sound and quality scientific presentation, presenting the concepts with clarity and precision. However, a few minor issues still exist, and with careful consideration and revision, these could be addressed to create an even better version. Here are some examples:
1, in figure 4A and 4B, for the bar charts, it is better to which one is 24h and which one is 48h.
2, In line 145-148, the author mentioned that “MPT0G236 increased the acetylation of α-tubulin and histone H3”, is this based on the cells should show same amount histone H3, including the acetylated one? the answer from the gel data seems to be no when combining the histone H3 and the acetyl histone H3. It looks like with higher dose treatment, the total protein expression of α-tubulin or histone H3 are higher too. In this case, the author should also take the effect of MPT0G236 on protein expression level into consideration. To deal with question, the author can considerate adding a set of figure of “the ratio of acetyl protein to non-acetyl protein”.
3, Figure 6A and 6B, the gel data, this has similar commons as above, as the dose increase, the total protein of caspase 8 (cleaved and non-cleaved) and caspase 9 (cleaved and non-cleaved) also increase. So this cleaved forms caspase 8 or 9 is all from the HDAC inhibitor treatment? Or is it also partially because of higher expression level of those proteins?
4, Line 185 – 185, corresponding to Figure 5, the author claimed that several protein expression level decreases in HTC-116, when MPT0G236 concentration is above 0.3 uM, however, when the concentration is at 0.1uM, the protein expression level are also higher than control group. This is very interesting, is this effect dose-dependent? The author should also explain this.
5, the author used a lot of data show that the new compound MPT0G236 works better that FDA proved SAHA, it would be great if the author can give more discussion about how and why? What is the difference between the two compounds? Why they work differently? This would need the author talk more about the mechanism about how HDAC inhibitor works on those enzymes. Here is one reference “Histone deacetylase inhibitors: molecular mechanisms of action” by P A Marks.
6, In the discussion section, particularly in lines 299-330, the author extensively explores how other HDAC inhibitors induce apoptosis. However, the relationship between MPT0G236 and apoptosis in this study remains somewhat unclear. The data presented only demonstrate an elevated level of cleaved caspase 8, caspase 9 and others, indicating potential involvement in apoptosis. However, the precise mechanism by which this HDAC inhibitor interacts with HDAC and subsequently triggers the downstream regulation of the caspase-dependent pathway remains insufficiently explained. Further clarification is needed to fully understand the workings of MPT0G236 in inducing apoptosis.
Author Response
Please find the Response to Reviewer 1 Comments as attached file.

Reviewer 2 Report
Dear Editor,
This manuscript evaluated the in vitro anti-tumor activity of the pan-HDAC inhibitor MPT0G236 in colorectal cancer cell lines.
Following are suggestions to strengthen the impact of the manuscript:
Major:
1. Can the authors explain the novelty of this manuscript from the previously published paper? (reference 22. Lee, H.Y.; Chang, C.Y.; Su, C.J.; Huang, H.L.; Mehndiratta, S.; Chao, Y.H.; Hsu, C.M.; Kumar, S.; Sung, T.Y.; Huang, Y.Z.; Li, 481 Y.H.; Yang, C.R.; Liou, J.P. 2-(Phenylsulfonyl)quinoline N-hydroxyacrylamides as potent anticancer agents inhibiting histone 482 deacetylase. Eur. J. Med. Chem. 2016, 122, 92–101.)
2. MTT and SRB assays are similar assay for cytotoxicity measurement although based on different mechanisms. What is the reasonale to use both methods here? How to explain the difference between IC50 and GI50 of the same compound in different assays?
Minor:
1. Figure1, how did the author get IC50 values from these connected data points? It’s better to fit the data into sigmoidal curves to get accurate IC50 values.
Table1, please provide individual IC50 curves in the supplementary.
Author Response
Please find the Response to Reviewer 2 Comments as attached file.

Reviewer 3 Report
The paper examines how the administration of MPY0G236, a pan-HDAC inhibitor, affects colorectal cancer cells (using two cell lines).
In addition to the HDAC inhibitory effect, the paper examines cell cycle analysis, associated protein expression, and apoptosis in the two cell lines. In addition, in all figures, various concentrations have been shaken and examined. It seems to me that a great deal of work has been done, but why not just decide on a single concentration based on the inhibition experiments? If this is the case, then the content of the study is simply an analysis of the cell cycle and apoptosis, and the investigation of other characteristics of cancer malignancy (observation of migration, proliferation, etc., which is standard), for example, the effect on cancer stem cells, if present, as well as some animal experiments, etc. experiments, etc... There are many experiments that need to be done.
-------------
Author Response
Please find the Response to Reviewer 3 Comments as attached file.

Round 2
Reviewer 3 Report
AUthors answered my issue.